# The Effect of the Security and Exchange Commission's Investigations into Corporate Social Responsibility Performance

Karel Hrazdil [1,*] , Jeong-Bon Kim [1] and Xin Li [2]

1   Beedie School of Business, Simon Fraser University, Vancouver, BC V5A 1S6, Canada; jeong_bon_kim@sfu.ca
2   School of Business, Trinity Western University, Langley, BC V2Y 1Y1, Canada; cynthia.li@twu.ca
*   Correspondence: karel_hrazdil@sfu.ca; Tel.: +1-778-782-6790

**Abstract:** We examine the effect of the Security and Exchange Commission's (SEC) investigations into firms' corporate social responsibility (CSR) performance. Adopting a staggered event study setting and analyzing all public and private SEC investigations into possible violations of federal securities laws, we find that firms reduce their investment in ESG-related activities and experience significantly lower CSR performance while being investigated by the SEC. This baseline finding is more pronounced among firms that appoint a large auditor or force their CEO to resign. To address concerns about potential endogeneity, we also conduct a multiperiod dynamic analysis and estimate our baseline regressions using the propensity-score-matched sample. Our results further reveal that the negative effect of SEC investigations on CSR performance manifests in CSR activities related to corporate governance and firms' products. Overall, we highlight some unintended consequences of SEC investigations.

**Keywords:** SEC investigations; CSR; ESG; corporate future orientation

## 1. Introduction

The Securities and Exchange Commission (SEC) in the United States was established with the mission of safeguarding the interests of investors, promoting the creation of capital, and upholding the integrity and efficiency of financial markets. Through its Division of Enforcement, the SEC investigates alleged breaches of securities law, such as unregistered securities offerings, accounting errors, insider trading, negligence, market manipulation, and various types of fraud. The investigation process is held in secrecy, with only a small group of people (i.e., the SEC staff, high-level managers of the company under investigation, and outside counsel) being involved in resolving the issue [1,2]. It is not mandatory for firms to disclose such SEC investigations, even when there is a possibility of an enforcement action; as a result, outside stakeholders, including shareholders, are often unaware of the investigations and only learn of the investigations and/or enforcement actions ex post facto.

Although SEC investigations are aimed at improving investor confidence and the integrity of capital markets, these investigations, which are of a secretive nature, also create significant information asymmetries between firms and stakeholders and impose significant economic consequences on the firms being investigated. Specifically, as key executives may update their beliefs about the likelihood that potential wrongdoing is detected, they may modify their actions once they are made aware of an SEC regulatory inquiry [3]. For example, prior literature documents that SEC investigations are associated with a substantial decline in future firm performance and severe insider trading [4], increased analyst coverage, an increased likelihood of appointing a large auditor, high litigation risk [5], decreased capital expenditures, and increased likelihoods of CEO turnover, earnings restatements, and class action lawsuits [3]. Although the SEC currently does not regulate CSR activities, failures related to the environmental, social, and governance (ESG) reporting and performance are investigated by the SEC and/or external auditors. (For

example, on 22 November 2022, Goldman Sachs Asset Management, L.P. was brought under scrutiny by the SEC due to lapses in its policies and procedures. These lapses pertained to two mutual funds and a separately managed account strategy marketed as an ESG investment. Over the period from April 2017 to February 2020, Goldman Sachs experienced a series of policy and procedural failures in relation to the ESG research utilized by its investment teams for selecting and monitoring securities. Furthermore, the firm lacked written policies and procedures for ESG research in one of its products, and even when policies and procedures were eventually established, they were not consistently adhered to. To settle the charges, Goldman Sachs agreed to pay a USD 4 million penalty (https://www.sec.gov/news/press-release/2022-209 (accessed on 27 August 2023))) However, empirical evidence on the relationship between SEC investigations and firm CSR performance is scarce.

As there is some probability that the SEC will detect misconduct and impose penalties on firms that are under investigation, we are the first to analyze whether managers of firms investigated by the SEC tend to change their actions with regard to CSR practices during periods of increased scrutiny. We postulate that SEC investigations can affect firms' CSR performance in two opposing directions. On the one hand, SEC investigations may lead to worse CSR performance for the following reasons. Most CSR activities are future-oriented in nature. As such, firms typically bear the costs of CSR activities immediately, but enjoy the benefits associated therewith over multiple periods or relatively long periods of time [6]. This implies that investment in CSR activities may worsen a firm's short-term earnings performance. In addition, SEC investigations generally increase the likelihood of a shareholder class action lawsuit for the dismissal of the CEO [3,7]. When firms face higher regulatory costs and managers face a higher likelihood of forced turnover, managers are likely to avoid investing in future-oriented activities that may not materialize during their tenure [8]. On the other hand, managers of firms investigated by the SEC may strategically increase their CSR performance as "window dressing", to reduce the likelihood or severity of enforcement actions, or to shorten the investigation period [9,10]. For example, Jia et al. (2019) [11] find that firms increased their CSR practices more when they were selected as pilot firms for an SEC regulation change, thereby suggesting that firms may use CSR to insure against stock price crash risk. Others, such as Tran and O'Sullivan (2020) [12] find that firms with higher CSR performance are less likely to be investigated by the SEC for financial misstatements because these firms are perceived to engage less in financial misstatements and the reputational effect of CSR activities on firm valuation reduces the likelihood of an SEC enforcement action.

Given the above competing explanations on the influence of SEC investigation on CSR performance, the directional effect of SEC investigations on CSR performance is unclear ex ante and ultimately an empirical question. To our knowledge, however, prior research has paid little attention to this question. To provide systematic evidence of the direction of this effect, we follow Blackburne et al. (2021a, 2021b; Blackburne and Quinn, 2023) [3–5] and utilize their novel dataset of all the SEC's Division of Enforcement investigations. This dataset covers investigations during the period between 2000 and 2017 and contains the opening and closing dates of the SEC investigations. Prior research focuses mainly on investigations that the SEC considers to be material and subject to enforcement (i.e., Accounting and Auditing Enforcement Releases (AAERs); [12–15]). However, such investigations represent only about 2% of all SEC investigations. Unlike prior research in this area, we analyze a much broader scope of SEC investigations in the CSR setting, which helps mitigate concerns about partial observability, SEC enforcement subjectivity, and timing errors [2,16] (For example, researchers must first determine which firms are presumed to engage in financial reporting misconduct before they can test which firm and managerial characteristics are associated with misconduct. The standard proxies (e.g., restatements, lawsuits, and AAERs) used in the literature are subject to partial observability, as the knowledge of financial misconduct comes almost exclusively from firms that have

been identified and prosecuted by the SEC, and the characteristics of these firms may exhibit dissimilar traits compared to those engaging in fraud that goes undetected [17]).

We take advantage of the staggered nature of SEC investigations and show that SEC investigations lead to a significant and substantial decrease in CSR performance during the investigation period. This finding supports our conjecture that improved regulations and/or heightened scrutiny associated therewith negatively affect firms' future-oriented investment and CSR performance. Our cross-sectional results further reveal that the negative effect of SEC investigations on CSR performance is more pronounced among firms audited by one of the Big Four auditors or firms with forced CEO turnovers.

We also conduct an array of additional robustness tests and address endogeneity concerns. First, we conduct a multiperiod dynamic analysis and exclude changes in CSR performance outside the frame of SEC investigations to control for possible improvements in CSR as a measure of firms' actions aimed at reducing the likelihood of being investigated by the SEC and other agencies. Second, we employ the propensity score matching (PSM) method to enhance the comparability between the companies undergoing treatment and those in the control group. Specifically, we identify a control firm comparable to each treatment firm in various dimensions by implementing one-to-one nearest neighbor matching without replacement [18,19]. Finally, we investigate how SEC investigations affect CSR performance along its seven dimensions and find that the negative effect of SEC investigations on CSR performance manifests itself in the areas related to "corporate governance and product".

This study makes several novel contributions to the literature. First, we contribute to the debate on the economic consequences of SEC investigations by exploring how SEC investigations affect a particular corporate behavior related to CSR performance. Although past literature documents how regulatory enforcements are associated with different managerial decisions and firms' activities (e.g., [15,20–22]), the primary focus has been on firms that were caught (by analyzing, e.g., AAERs, restatements, and lawsuits), which are publicly announced and represent only a small fraction of all SEC investigations. We follow a recent stream of regulation literature and extend its scope to include all SEC investigations into different forms of violations that can be triggered by various impulses, such as whistleblowing, press reports, or surveillance activities [3,5]. We explore how SEC investigations affect managerial decisions concerning future-oriented benefits such as ESG investments. To our knowledge, our study is the first to document that such investigations predict material declines in CSR performance. Second, unlike prior studies that analyze the determinants of SEC investigations (e.g., [12,15,22]), which are subject to extensive criticism for their data limitations [3,17], we utilize all SEC investigations as the treatment variable to examine how the ex ante level of regulation impacts firms' subsequent CSR performance. Our results provide novel evidence that SEC investigations have unintended consequences on firms' CSR performance, irrespective of their severity.

Our study has implications not only for the literature on regulatory enforcement and financial reporting misconduct but also broader implications for accounting standards. As investors demand information on firms' impact on climate and other ESG-related activities, the SEC is responding by proposing new reporting and disclosure rules that would mandate detailed disclosure of climate-related risks and greenhouse gas (GHG) emissions, facilitate comparable ESG disclosures, and thus ensure that statements made to investors are not false or misleading.

This paper proceeds as follows. Section 2 presents the related literature and develops the hypotheses. Section 3 describes our sample and research design. Section 4 presents the main empirical results and sensitivity analysis, and Section 5 concludes the paper.

## 2. Literature and Hypotheses

The Division of Enforcement (DoE) of the SEC recommends and carries out investigations into potential securities law violations, such as the misrepresentation or omission of material information, manipulation of the market prices of securities, insider trading, or

other fraudulent acts in the issuance of or reporting on public securities. If the DoE makes a preliminary determination from their investigation and intends to recommend that the SEC pursue an enforcement action, the investigation typically enters the "Wells process" stage, during which the SEC notifies the firm of its intent to pursue an enforcement action (i.e., a Wells notice). The firm subject to the alleged violation has an opportunity to submit a response (i.e., a Wells submission), after which the DoE makes its recommendation to the SEC and the Commission votes on whether to authorize an enforcement action. The investigation period—that is, the time between the start of an investigation and the initiation of the first enforcement action—is about two years, but this period can be longer in certain cases [5]. If the Commission votes to approve the DoE's recommendations, it will commence with an enforcement action. The SEC has successfully obtained judgment on over 90% of its enforcement actions [23].

A large body of literature analyzes known financial reporting misconduct and focuses mainly on firms that ultimately received enforcement actions such as AAERs, as the outcome of SEC investigations. For example, Beneish (1999) [13] and Dechow et al. (2011) [14] use financial statement variables to predict AAERs with what is now known as the M-Score and F-Score, respectively. Other authors analyze nonfinancial measures [24], control for firm CSR performance [12,15] and apply machine learning to analyze financial statements and MD&A sections [25,26] to detect fraudulent activities. However, due to data limitations, we know little about the SEC's investigative procedures, as these procedures are held in confidence, and exemptions granted to law enforcement under the Freedom of Information Act empower the SEC to retain a considerable volume of information.

Building on the prior literature on SEC enforcement actions relating to financial reporting misconduct, Blackburne et al. (2021a; 2021b) [3,4] utilize a novel dataset that contains 12,861 SEC investigations during the period from 2000 to 2017. Compared to AAERs and shareholder class action lawsuits, this dataset consists of a much broader range of SEC investigations, and covers investigations including those related to fraud, foreign corrupt practices, market manipulation, broker–dealer violations, and securities trading. According to Blackburne et al. (2021b) [4], the SEC does not provide further details of each investigation, including the dates of communication with the entity under investigation and outcomes of each investigation.

Recent literature analyzes all SEC investigations and provides new insights into their consequences for managerial behavior, even for firms with non-AAER outcomes (i.e., non-severe SEC investigations). First, Blackburne et al. (2021b) [4] find that SEC investigations predict material economic declines in future firm performance, and that this information (i.e., the undisclosed nature of these investigations) is exploited by corporate insiders for personal gain. Second, Blackburne and Quinn (2023) [5] analyze the determinants of the disclosure speed of SEC investigations and document that external monitoring and litigation risk are associated with about 100% and 40% faster disclosure, respectively, whereas managerial entrenchment is associated with about 30% slower disclosure. Finally, Blackburne et al. (2021a) [3] document a "regulatory observer" effect of SEC investigations, whereby managers perceive SEC actions to be more punitive if they employ discretionary reporting and improve accounting misstatement risk, reduce accounting irregularities, and increase conservatism. The authors further suggest that firms investigated by the SEC exhibit decreased capital and R&D expenditures and increased likelihoods of CEO turnover, earnings restatements, and class action lawsuits, regardless of whether an enforcement action is pursued. We contribute to this literature by exploring whether such investigations predict material changes in CSR performance.

The growing public awareness of CSR practices drives firms to increase their investment in CSR-related activities. In some cases, companies may feel pressure to improve their CSR practices as a result of negative publicity or the reputational damage caused by an SEC investigation. This can lead to changes such as increased transparency and disclosure, improved environmental or labor practices, or better governance. However, it is also possible that companies may make superficial or cosmetic changes to their CSR

practices to appear more compliant or avoid further regulatory scrutiny without making any substantive changes. As CSR activities are costly and firms have limited resources to allocate to such activities, additional costs associated with responding to SEC investigations may further divert resources away from CSR engagements.

Prior literature documents a variety of reasons why firms engage in CSR. Apart from the social benefits [27,28] of being a responsible corporate entity, a firm also strategically manages its CSR activities to mitigate the conflicts between shareholders and other stakeholders [29,30], to enhance its reputation or for window-dressing purposes [9,10], to improve its competitive advantage [31,32] or to retain its employees [19,33]. The nonpublic nature of SEC investigations provides a unique setting in which to understand how regulatory scrutiny affects managerial behavior with regard to CSR, as the typical external stakeholders are unaware of the alleged misconduct. Once the SEC initiates its investigation, managers may update their beliefs about the likelihood that potential wrongdoing will be detected, and their career considerations may influence their actions during the investigation process [3]. As a result, SEC investigations can impact firms' CSR performance during the period of increased scrutiny in the following ways.

On the one hand, managers of firms facing a greater threat of regulatory enforcement may strategically increase their investment in future-oriented CSR activities to window-dress, reduce the likelihood or severity of an ensuing enforcement action, or shorten the investigation period [9,10,34]. Consistent with this view, Leone et al. (2021) [35] find that when a firm under investigation cooperates in good faith, the likelihood of SEC enforcement actions and monetary penalties decreases; Marquis and Qian (2014) [36] show that greater government monitoring leads to more substantive CSR reports; Tran and O'Sullivan (2020) [12] find that firms with higher CSR performance are less likely to be investigated by the SEC for financial misstatements; and Tsang et al. (2023) [37] find that once a US-listed foreign firm's home country signs the Multilateral Memorandum of Understanding Concerning Consultation and Cooperation and the Exchange of Information (MMoU), the firm increases its voluntary disclosure, due to greater concerns about regulatory enforcement and an increased investor demand for information. In addition, improving stakeholder engagement through CSR performance may have value as "insurance" that helps to reduce the regulatory threat and reduce CEO turnover risk [38,39]. On the other hand, SEC investigations may lead to a decrease in CSR-related investments. As a future-oriented activity, CSR incurs short-term costs and delays long-term benefits [6]. As firms face higher regulatory risk and managers face a higher likelihood of job turnover, irrespective of whether the SEC imposes a penalty, managers tend to avoid investing in future-oriented activities that may not materialize during their tenure [8].

Finally, managers may not adjust their CSR-related investments once they become aware of an SEC investigation if they believe that their firm has not committed misconduct, that external monitors are not aware of the alleged misconduct, or that any changes in behavior during the investigation will be viewed as an admission of guilt [5]. Furthermore, a firm may not be aware of the precise nature or severity of the investigation unless the SEC issues a Wells notice.

Given the contending predictions above, the directional effect of SEC investigations on CSR performance is ultimately an empirical question. To provide systematic evidence of this unexplored issue, we test our first hypothesis below, stated in null form.

**H1.** *Ceteris paribus, SEC investigations have no impact on firms' CSR performance during the investigation period.*

We further examine two theoretical mechanisms or channels through which SEC investigations may potentially affect CSR practices: the engagement of a large auditor and forced CEO turnover. First, external auditing provides reasonable assurance of the quality of corporate disclosure and is an effective control mechanism with which to monitor managers [40,41]. External auditors are expected to constrain opportunistic behaviors as well as reduce informational risk [42]. According to agency theory, external auditors reduce

the information asymmetry between managers and investors, which improves resource allocation and contracting efficiency [40]. This theory further posits that managers of companies known for their transparency can face significant repercussions for withholding information [43]. In cases where auditors become aware that managers are reluctant to disclose details about an SEC investigation, whether due to leaks or delayed disclosure, auditors may be more inclined to issue a qualified opinion.

Past literature documents the fact that large auditors with a well-known brand name (i.e., the Big Four) deliver higher quality audits because they exert more effort [44,45], face higher reputational and litigation risk in the event of a misreporting problem [46], charge higher fees [47,48], and deploy more human capital and technology resources [49]. Furthermore, past studies suggest that firms audited by Big Four auditors exhibit faster information disclosure [50,51], especially during periods of SEC investigation [5].

Generally, higher-quality CSR disclosures are associated with better CSR performance [52]; however, firms often use CSR reporting as a strategic device for window dressing [53]. Accordingly, Chen et al. (2016) [54] argue that allocating greater resources to enhance the quality of audits enhances the credibility of voluntary CSR reports and makes these reports more informative for investors. Therefore, selecting a Big Four auditor plays an important role in enhancing the reliability and credibility of the disclosed CSR information. As there are two potential forces that affect CSR performance (i.e., SEC investigation and selecting a Big Four auditor), we posit that the effect of an SEC investigation on a firm's CSR is amplified when a firm is audited by a Big Four auditor.

**H2.** *Ceteris paribus, the effect of an SEC investigation on firms' CSR performance during the investigation period is more pronounced among firms audited by a Big Four auditor.*

Second, the strategic motivation for CSR activities remains a debatable issue. Hambrick and Mason's (1984) [55] and Hambrick's (2007) [56] upper echelons theory predicts that organizational outcomes reflect the values and cognitive biases of the upper echelons of an organization (i.e., corporate executives), who exert a significant influence on corporate policies and activities. According to upper echelons theory, strategic choices concerning CSR activities are shaped not only by situational characteristics (e.g., the external environment) and/or organizational performance, but are also influenced by manager characteristics [57,58], such that CEOs play an important role in shaping firms' CSR performance [59,60].

Prior literature further suggests that both financial and CSR performance are negatively related to CEO turnover (e.g., [61,62]). Solomon and Soltes (2021) [2] show that among firms under investigation, CEOs who disclose the investigation are more likely to experience turnover. Accordingly, we posit that the negative relation between SEC investigations and CSR is more pronounced when firms force their CEO to resign.

**H3.** *Ceteris paribus, the effect of an SEC investigation on firms' CSR performance during the investigation period is more pronounced when firms force their CEO to resign.*

## 3. Data and Methods

We obtain SEC investigation data from Blackburne et al. (2021b) [4], who developed a comprehensive dataset of all active SEC investigations during the period of 2000–2017, through formal requests and direct communications with the Freedom of Information Act office (FOIA) of the SEC. This dataset consists of 12,861 observations and covers exchanged-listed companies, registered investment advisers, broker-dealers, mutual funds, exchange-traded funds, and other entities under the Commission's purview [4]. The types of SEC investigations go beyond GAAP violations and contain a variety of issues related to financial fraud, foreign corrupt practices, market manipulation, broker-dealer violations, and securities trading.

We further obtain CSR performance data from the MSCI ESG KLD STATS database (hereafter "KLD"). KLD's (produced by Kinder, Lydenberg, and Domini) CSR ratings have been widely used in prior literature (e.g., [19,63–66]) and provide binary summaries of

positive and negative CSR ratings (i.e., strengths and concerns) along seven dimensions related to community, corporate governance, diversity, employee relations, the environment, human rights, and products. We utilize the KLD database, as it is widely recognized as the most extensive, publicly available, and largest multidimensional corporate social performance database and the prevailing standard in current CSR research. While raters employ diverse methods and variables to assess the same construct, there exists a substantial consensus regarding the elements of social responsibility, with comparable coverage of overarching themes, including environmental and social performance [67]. Prior research has also established a strong correlation between KLD environmental strengths and AS-SET4 and GES EP ratings. They align closely in evaluating the performance construct for companies listed on the US MSCI World Index during the period from 2003 to 2011 [68].

We calculate the overall CSR performance (*CSR_net*) as the overall positive CSR performance (*CSR_str*, or the sum of all CSR strengths) minus the overall negative CSR performance (*CSR_con*, or the sum of all CSR concerns). We also obtain variables related to firm characteristics from Compustat and institutional ownership data from Thompson Reuters databases. We match KLD data with firms' financial data from Compustat and Thompson Reuters and drop the post-2017 data, because our sample period is limited by the availability of investigation data. After excluding firms with missing data for the dependent and independent variables, the final sample consists of 36,111 firm-year observations during the period 1999–2017. We winsorize all continuous variables at their 1st and 99th percentiles to exclude the effect of outliers.

To examine the relation between SEC investigations and firms' CSR engagement, we follow the previous literature (e.g., [69–71]) and use a set of basic regression models with one of the CSR variables as the dependent variable and the SEC investigation as the independent variable. Equation (1) summarizes the model. All variables used in Equation (1) are as defined in the next section.

$$CSR_{i,t} = \alpha_0 + \beta_1 Investigation_{i,t} + \beta X_{i,t} + \mu_i + \mu_t + \varepsilon_{i,t} \tag{1}$$

*Variable Measurement*

1.  Dependent variable: Following Blackburne et al. (2021b) [4], we define the binary variable *Investigation* based on their dataset, which is constructed through formal requests and direct communications with the FOIA office of the SEC. *Investigation* is an indicator variable that equals one if the company is under SEC investigation in a certain fiscal year and zero otherwise—that is, before and after the investigation and for all other companies that have never been the subject of an SEC investigation. The coefficient on the key variable of our interest (i.e., $\beta_1$) measures the effect of SEC investigations on firms' CSR engagement. Hypothesis $H1_0$ tests whether $\beta_1$ is different from zero.

2.  Independent variables: Following recent CSR literature [19,72,73], we include a set of control variables, where *X* is a vector of firm-specific characteristics such as total assets (*TA*), Tobin's Q (*TQ*), losses (*Loss*), leverage (*LEV*), profitability (*Profit*), tangibility (*Tangible*), cash holdings (*Cash*), and institutional ownership (*IO*). The indicator variables $\mu_t$ and $\mu_j$ refer to year and firm fixed effects, respectively.

We cluster the standard errors at the firm level to correct for serial correlation within a firm. Table 1 provides a list of detailed definitions of all variables used in this paper.

**Table 1.** Detailed definition of main variables.

| Variables | Description |
|---|---|
| *CSR_con* | CSR concerns, based on KLD ESG ratings |
| *CSR_str* | CSR strengths, based on KLD ESG ratings |
| *CSR_net* | Net CSR performance, constructed as *CSR_str* minus *CSR_con* |
| $\Delta CSR\_net$ | Change in overall CSR performance compared to the previous year |
| $\Delta CSR\_str$ | Change in positive CSR performance compared to the previous year |
| $\Delta CSR\_con$ | Change in negative CSR performance compared to the previous year |
| *Investigation* | An indicator variable that equals one when a firm is under investigation during the year and zero otherwise (based on the SEC investigation dataset of Blackburne et al., 2021b [4]) |
| *Investigation* − 1 | An indicator variable that equals one in one year before the start of SEC investigation and zero otherwise |
| *Investigation* + 1 | An indicator variable that equals one in one year after the SEC concludes investigation of the firm and zero otherwise |
| *TA* | Natural logarithm of total assets in year *t* |
| *TQ* | Tobin's Q, calculated as the market value of equity plus the book value of debt (calculated as long-term debt plus short-term debt), scaled by the book value of total assets in year *t* |
| *Loss* | An indicator variable that equals one if income before extraordinary items is negative and zero otherwise |
| *LEV* | Leverage ratio, calculated as short-term debt plus long-term debt scaled by total assets in year *t* |
| *Profit* | Profitability, calculated as operating income before depreciation divided by total assets in year *t* |
| *Tangible* | Total property, plant, and equipment (PP&E) over total assets in year *t* |
| *Cash* | Cash and short-term investments over total assets in year *t* |
| *IO* | The percentage of shares owned by institutions in year *t* |
| *BIG4* | An indicator variable that equals one if a firm's auditor is one of the Big 4 audit firms during the fiscal year and zero otherwise |
| *Forced* | An indicator variable that equals one if a firm's CEO was forced to quit (source: Peters and Wagner 2014 [74]; https://www.florianpeters.org/data/ (accessed on 27 August 2023)) and zero otherwise |

## 4. Descriptive Statistics and Results

Table 2 reports the yearly distribution of our sample to indicate the number of firms that are investigated by the SEC each year. The total number of firm-year observations is 36,111, wherein the SEC investigates on average 10.47% (3783 out of 36,111) of the publicly listed firms. The dramatic increase in the number of firms in 2003 is due to the inclusion of small-cap US companies and the Broad Market Social Index in the KLD database starting in 2003.

**Table 2.** Yearly distributions.

| Fiscal Year | Investigation | | Total |
|---|---|---|---|
| | **No** | **Yes** | |
| 1999 | 473 | 23 | 496 |
| 2000 | 455 | 31 | 486 |
| 2001 | 793 | 45 | 838 |
| 2002 | 789 | 64 | 853 |
| 2003 | 2074 | 206 | 2280 |
| 2004 | 2062 | 293 | 2355 |
| 2005 | 1954 | 327 | 2281 |
| 2006 | 1835 | 412 | 2247 |
| 2007 | 1906 | 359 | 2265 |
| 2008 | 1950 | 359 | 2309 |
| 2009 | 1997 | 311 | 2308 |

**Table 2.** *Cont.*

| Fiscal Year | Investigation | | Total |
|---|---|---|---|
| | **No** | **Yes** | |
| 2010 | 2103 | 284 | 2387 |
| 2011 | 1974 | 226 | 2200 |
| 2012 | 1972 | 233 | 2205 |
| 2013 | 2144 | 193 | 2337 |
| 2014 | 1918 | 163 | 2081 |
| 2015 | 1819 | 128 | 1947 |
| 2016 | 1861 | 87 | 1948 |
| 2017 | 2249 | 39 | 2288 |
| Total | 32,328 | 3783 | 36,111 |

Table 3 presents the descriptive statistics for all variables used in Equation (1) (*n* = 36,111). The mean net CSR score is –0.07, with a median of zero, which shows a symmetric distribution among CSR ratings. The mean *Investigation* is 0.10, which corresponds with the result in Table 2 that show 10.47% firm-years are under SEC investigation.

**Table 3.** Descriptive Statistics.

| Variable | Mean | S.D. | Min. | 25% | Median | 75% | Max. |
|---|---|---|---|---|---|---|---|
| *CSR_net* | –0.07 | 2.48 | –12.00 | –1.00 | 0.00 | 0.00 | 19.00 |
| *CSR_str* | 1.53 | 2.36 | 0.00 | 0.00 | 0.00 | 1.00 | 22.00 |
| *CSR_con* | 1.60 | 1.86 | 0.00 | 0.00 | 1.00 | 2.00 | 18.00 |
| *Investigation* | 0.10 | 0.31 | 0.00 | 0.00 | 0.00 | 0.00 | 1.00 |
| *TA* | 7.46 | 1.76 | 3.39 | 6.17 | 7.35 | 8.57 | 13.47 |
| *TQ* | 1.48 | 1.53 | 0.01 | 0.51 | 1.01 | 1.85 | 14.99 |
| *Loss* | 0.21 | 0.41 | 0.00 | 0.00 | 0.00 | 0.00 | 1.00 |
| *LEV* | 2.29 | 4.19 | –19.30 | 0.51 | 1.13 | 2.41 | 35.83 |
| *Profit* | 0.02 | 0.14 | –1.14 | 0.01 | 0.04 | 0.08 | 0.29 |
| *Tangible* | 0.24 | 0.24 | 0.00 | 0.05 | 0.15 | 0.36 | 0.91 |
| *Cash* | 0.18 | 0.21 | 0.00 | 0.03 | 0.10 | 0.25 | 0.94 |
| *IO* | 0.68 | 0.27 | 0.00 | 0.51 | 0.73 | 0.88 | 1.27 |

*4.1. Baseline Results*

Models (1)–(3) in Table 4 reveal how SEC investigations affect firms' CSR performance (*CSR_net*) and its components (*CSR_str* and *CSR_con*) without the control variables, whereas Models (4)–(6) display the results with a set of control variables. All regressions further include year and firm fixed effects. As shown in Table 4, the coefficients on *Investigation* are significantly negative in Models (1) and (4), thus indicating that SEC investigations have a negative impact on firms' overall CSR performance. (Inferences are similar (although less significant) when we only include firms that were investigated during the sample period (results not tabulated)). Of the two CSR components, the coefficients on *Investigation* are positive, larger in magnitude and statistically significant for *CSR_con* (0.294 and 0.274 in Models (3) and (6), respectively) compared to the statistically nonsignificant coefficients for *CSR_str* (0.085 and 0.072 in Models (2) and (5), respectively). These results indicate that SEC investigations exacerbate CSR concerns, which is consistent with CSR being negatively associated with SEC investigations (As a sensitivity analysis, we investigate whether more severe SEC investigations (such as AAERs) have a stronger impact on CSR performance. While one might expect that managers reduce investments in CSR even more during more severe investigations, we find that the increased regulatory scrutiny associated with non-severe investigations is sufficient for a significant reduction in CSR investment (results not tabulated)).

**Table 4.** Baseline Results.

| Model | (1) | (2) | (3) | (4) | (5) | (6) |
|---|---|---|---|---|---|---|
| Variable | *CSR_net* | *CSR_str* | *CSR_con* | *CSR_net* | *CSR_str* | *CSR_con* |
| *Investigation* | −0.209 *** | 0.085 | 0.294 *** | −0.202 *** | 0.072 | 0.274 *** |
| | [0.003] | [0.138] | [0.000] | [0.004] | [0.206] | [0.000] |
| *TA* | | | | −0.030 | 0.263 *** | 0.294 *** |
| | | | | [0.594] | [0.000] | [0.000] |
| *TQ* | | | | −0.035 * | −0.024 | 0.011 |
| | | | | [0.081] | [0.169] | [0.333] |
| *Loss* | | | | −0.105 ** | −0.045 | 0.060 * |
| | | | | [0.015] | [0.159] | [0.066] |
| *LEV* | | | | 0.006 | 0.001 | −0.005 |
| | | | | [0.229] | [0.813] | [0.138] |
| *Profit* | | | | 0.056 | −0.121 | −0.177 * |
| | | | | [0.678] | [0.235] | [0.054] |
| *Tangible* | | | | 0.962 ** | 0.544 * | −0.418 |
| | | | | [0.016] | [0.058] | [0.106] |
| *Cash* | | | | 0.454 ** | 0.361 ** | −0.094 |
| | | | | [0.014] | [0.021] | [0.440] |
| *IO* | | | | −0.338 *** | −0.218 ** | 0.120 |
| | | | | [0.004] | [0.012] | [0.104] |
| Year FE | Yes | Yes | Yes | Yes | Yes | Yes |
| Firm FE | Yes | Yes | Yes | Yes | Yes | Yes |
| Clustered by Firm | Yes | Yes | Yes | Yes | Yes | Yes |
| N | 36,111 | 36,111 | 36,111 | 36,111 | 36,111 | 36,111 |
| Adjusted $R^2$ | 0.542 | 0.695 | 0.678 | 0.543 | 0.696 | 0.681 |

Table 1 provides definitions for all variables, with continuous variables being winsorized at the 1st and 99th percentiles. *p*-values are enclosed in parentheses and are computed using robust standard errors. Significance levels are indicated by ***, **, and * for *p*-values less than 0.01, 0.05, and 0.10, respectively.

The effects of control variables on CSR performance are displayed in Models (4)–(6). We find that *Tangible* and *Cash* are significantly and positively associated with firms' overall CSR performance, whereas *TQ*, *Loss*, and *IO* are negatively associated with *CSR_net*. These results are consistent with those of prior studies. For example, firms with more cash on hand or higher tangible assets such as property, plant and equipment (PP&E) tend to have higher CSR performance, as higher levels of PP&E indicate a greater supply of internal resources and a higher ability to attract investment from external sources [19,75]. Furthermore, while some authors document a positive relationship between CSR and corporate financial performance (CFP) [76], others document a more limited magnitude of this effect and the causal relationship [77], and some find both over- and under-investment in CSR to be negatively related to a firm's short-term financial performance [78]. Finally, while some document that institutional ownership is a concave function of CSR [79,80], others argue that as shareholdings of institutional investors become larger, they can shape the link between CSR involvement and firm value. This influence stems from the ability of institutional investors to encourage firms to trim excessive CSR spending through effective monitoring, ultimately resulting in a negative correlation between institutional ownership and CSR engagement [81,82].

*4.2. Cross-Sectional Results*

In cross-sectional studies, we first investigate how the presence of a Big Four auditor affects the baseline relation observed between SEC investigations and CSR. We use the indicator variable *BIG4* to identify firms that are audited by one of the Big Four auditors (KMPG, E&Y, PWC, and Deloitte) and include *BIG4* and the interaction term *Investigation*Big4* in the baseline model. In the second cross-sectional test, we examine how CEO turnover affects the baseline relation observed between SEC investigations and CSR. To investigate the direct link between firm performance and CEO turnover, we follow Jenter and Kanaan

(2015) [62] and focus on a strictly defined "forced" CEO turnover, which excludes reasons such as death, poor health, and the acceptance of another position internally, as well as CEO turnover associated with mergers and spin-offs, from our analysis. We use the indicator variable *Forced* to identify firms that experience forced CEO turnover and include *Forced* and the interaction term *Investigation\*Forced* in the baseline model.

Models (1)–(3) in Table 5 test whether and how the presence of a Big Four auditor affects the relation between SEC investigations and CSR performance. Model (1) shows that the coefficient on the interaction term *Investigation\*BIG4* is negative and significant (–0.284) for overall CSR performance (*CSR_net*). This finding suggests that firms audited by Big 4 auditors experience a further decrease in CSR performance compared to non-Big Four audited firms when faced with SEC investigations. The result for net CSR performance is driven by CSR concerns, where the interaction coefficient is positive and significant. These results are consistent with our second hypothesis and consistent with a plausibility that firms audited by large auditors disclose information pertaining to SEC investigations faster, which leads to a further decrease in CSR (especially that relating to environmental threats). The coefficient on *BIG4* in Model (1) is positive and significant, which suggests that firms with high CSR performance are audited by one of the Big Four firms, whereas the negative coefficient on *BIG4* in Model (3) suggests that large auditors shy away from firms with CSR concerns.

**Table 5.** Cross-sectional Results.

| Model | (1) | (2) | (3) | (4) | (5) | (6) |
|---|---|---|---|---|---|---|
| **Variable** | *CSR_net* | *CSR_str* | *CSR_con* | *CSR_net* | *CSR_str* | *CSR_con* |
| *Investigation\*BIG4* | −0.284 * | 0.0250 | 0.309 *** | | | |
| | [0.069] | [0.818] | [0.009] | | | |
| *BIG4* | 0.341 *** | 0.161 ** | −0.180 *** | | | |
| | [0.000] | [0.012] | [0.004] | | | |
| *Investigation\*Forced* | | | | −0.356 * | −0.081 | 0.276 * |
| | | | | [0.090] | [0.644] | [0.070] |
| *Forced* | | | | −0.011 | 0.026 | 0.037 |
| | | | | [0.906] | [0.722] | [0.539] |
| *Investigation* | 0.056 | 0.050 | −0.006 | −0.129 | 0.127 * | 0.256 *** |
| | [0.685] | [0.578] | [0.956] | [0.126] | [0.071] | [0.000] |
| Controls | Yes | Yes | Yes | Yes | Yes | Yes |
| Year FE | Yes | Yes | Yes | Yes | Yes | Yes |
| Firm FE | Yes | Yes | Yes | Yes | Yes | Yes |
| Clustered by Firm | Yes | Yes | Yes | Yes | Yes | Yes |
| N | 34,266 | 34,266 | 34,266 | 23,246 | 23,246 | 23,246 |
| Adjusted $R^2$ | 0.537 | 0.693 | 0.680 | 0.543 | 0.700 | 0.704 |

Table 1 provides definitions for all variables, with continuous variables being winsorized at the 1st and 99th percentiles. *p*-values are enclosed in parentheses and are computed using robust standard errors. Significance levels are indicated by \*\*\*, \*\*, and \* for *p*-values less than 0.01, 0.05, and 0.10, respectively. The controls include the same set as in Table 4.

Models (4)–(6) in Table 5 test whether forced CEO turnover affects the baseline relation observed between SEC investigations and CSR performance. Model (4) shows that the coefficient on the interaction term *Investigation\*Forced* is significantly negative (–0.356) for overall CSR performance. This result suggests that firms that experience forced CEO turnover tend to further decrease their CSR performance compared to their peers while being investigated by the SEC. These results support our third hypothesis and are consistent with forced CEO turnover being associated with poor CSR performance.

### 4.3. Endogeneity Concerns

We further conduct two sets of analyses to address potential endogeneity concerns. First, we conduct a multiperiod dynamic analysis to rule out the possibility that CSR performance decreases prior to or after SEC investigations. We include the year of the SEC investigation period (*Investigation*), together with the year prior to an investigation (*Investigation* − 1) and the year after an investigation (*Investigation* + 1) in the baseline model. Table 6 shows the results of the dynamic analysis, where only the coefficient on *Investigation* is significantly negative at the significance level of 1% (columns 1 and 3), while the coefficients on *Investigation* − 1 are statistically nonsignificant across all cases, and those on *Investigation* + 1 are nonsignificant (columns 1 and 2) and marginally significant at the 1% level (column 3). These results indicate that the decrease in CSR manifests only during the period of SEC investigations (or thereafter), suggesting that our baseline results in Table 4 are unlikely to be driven by a pre-existing parallel trend in the pre-investigation period.

**Table 6.** Dynamic Analysis of SEC Investigations.

| Model | (1) | (2) | (3) |
|---|---|---|---|
| Variable | *CSR_net* | *CSR_str* | *CSR_con* |
| *Investigation* − 1 | −0.054 | −0.003 | 0.051 |
| | [0.479] | [0.958] | [0.317] |
| *Investigation* | −0.218 *** | 0.080 | 0.299 *** |
| | [0.008] | [0.228] | [0.000] |
| *Investigation* + 1 | −0.075 | 0.071 | 0.145 *** |
| | [0.392] | [0.304] | [0.010] |
| Controls | Yes | Yes | Yes |
| Year FE | Yes | Yes | Yes |
| Firm FE | Yes | Yes | Yes |
| Clustered by Firm | Yes | Yes | Yes |
| N | 36,111 | 36,111 | 36,111 |
| Adjusted $R^2$ | 0.543 | 0.696 | 0.681 |

Table 1 provides definitions for all variables, with continuous variables being winsorized at the 1st and 99th percentiles. *p*-values are enclosed in parentheses and are computed using robust standard errors. Significance levels are indicated by *** for *p*-values less than 0.01, respectively. The controls include the same set as in Table 4.

Second, we conduct propensity score matching (PSM) by comparing the CSR performance of firms that experience SEC investigations (the treatment group) with the one-to-one matched firms that experience no investigations (the control group). The one-to-one matched control group is constructed based on the same set of control variables from the main test (i.e., *TA*, *TQ*, *LEV*, *Loss*, *Profit*, *Tangible*, *Cash*, and *IO*) with no replacement. We run the baseline model based on the sample constructed from the one-to-one propensity score matching analysis with 7,498 observations.

The results in Table 7 demonstrate that the coefficient on *Investigation* for *CSR_net* is negative and significant, which is consistent with the main results showing that CSR decreases when firms are investigated by the SEC. This finding suggests that our baseline results in Table 4 are unlikely to be driven by endogeneity problems associated with potential differences in known covariates between the treatment and control groups. The covariate test (untabulated) indicates statistical insignificance of the differences in the means between the treatment and control group, which demonstrates that the PSM yields a balanced sample. Overall, the results from the dynamic analysis and the propensity score matching tests lend further support to the causal relation between SEC investigations and firms' CSR performance.

**Table 7.** PSM Regressions.

| Model | (1) | (2) | (3) |
| --- | --- | --- | --- |
| **Variable** | *CSR_net* | *CSR_str* | *CSR_con* |
| *Investigation* | −0.214 ** | 0.114 * | 0.328 *** |
| | [0.012] | [0.061] | [0.000] |
| Controls | Yes | Yes | Yes |
| Year FE | Yes | Yes | Yes |
| Industry FE | Yes | Yes | Yes |
| Clustered by Industry | Yes | Yes | Yes |
| N | 7,498 | 7,498 | 7,498 |
| Adjusted $R^2$ | 0.209 | 0.435 | 0.424 |

Table 1 provides definitions for all variables, with continuous variables being winsorized at the 1st and 99th percentiles. Industry fixed effects (FE) are based on SIC 2-digit industries. *p*-values are enclosed in parentheses and are computed using robust standard errors. Significance levels are indicated by ***, **, and * for *p*-values less than 0.01, 0.05, and 0.10, respectively. The controls include the same set as in Table 4.

*4.4. Dimensions of CSR*

As a final test, we further re-estimate our baseline specification by partitioning the KLD ratings index into eight distinct ESG dimensions. The outcomes of this analysis are presented in Table 8. We use the seven aspects of the KLD index—corporate governance (*CGOV*), community (*COM*), diversity (*DIV*), employee relations (*EMP*), environment (*ENV*), human rights (*HUM*), and product (*PRO*)—as the dependent variables, respectively. The results are presented in columns (1)–(7). The coefficients on *Investigation* are negative and significant in models where *CGOV* and *PRO* are included as independent variables (with values of –0.147 and –0.033, respectively), and positive and significant in models where *DIV* is the independent variable (with a value of 0.063). These results indicate that SEC investigations significantly lower firms' CSR investments in the corporate governance and product categories and increase their investments in the diversity category. The two most significant dimensions (i.e., CGOV and DIV) are, to a large extent, related to corporate governance, suggesting that the effect of SEC investigations on CSR performance manifests through corporate governance dimensions rather than environmental or social dimensions. SEC investigations typically involve thorough examinations of financial reporting, disclosure methods, and adherence to securities laws and regulations. This heightened regulatory scrutiny can prompt a company under investigation to adjust its corporate governance practices. Similarly, an SEC investigation can have an impact on an organization's diversity and inclusion initiatives if allegations or evidence of discriminatory practices in hiring, promotion, compensation, or workplace culture are revealed.

**Table 8.** The Effect of SEC Investigations and CSR Performance Across Seven CSR Dimensions.

| Model | (1) | (2) | (3) | (4) | (5) | (6) | (7) |
| --- | --- | --- | --- | --- | --- | --- | --- |
| **Variable** | *CGOV* | *COM* | *DIV* | *EMP* | *ENV* | *HUM* | *PRO* |
| *Investigation* | −0.147 *** | 0.001 | 0.063 ** | −0.043 | −0.026 | −0.014 | −0.033 * |
| | [0.000] | [0.929] | [0.049] | [0.151] | [0.330] | [0.192] | [0.060] |
| Controls | Yes | Yes | Yes | Yes | Yes | Yes | Yes |
| Year FE | Yes | Yes | Yes | Yes | Yes | Yes | Yes |
| Firm FE | Yes | Yes | Yes | Yes | Yes | Yes | Yes |
| Clustered by Firm | Yes | Yes | Yes | Yes | Yes | Yes | Yes |
| N | 36,111 | 36,111 | 36,111 | 36,111 | 36,111 | 36,111 | 36,111 |
| Adjusted $R^2$ | 0.374 | 0.410 | 0.594 | 0.461 | 0.486 | 0.348 | 0.476 |

Table 1 provides definitions for all variables, with continuous variables being winsorized at the 1st and 99th percentiles. *p*-values are enclosed in parentheses and are computed using robust standard errors (clustered by state). Significance levels are indicated by ***, **, and * for *p*-values less than 0.01, 0.05, and 0.10, respectively. The controls include the same set as in Table 4.

## 5. Conclusions

SEC investigations are conducted in secrecy and cover a much broader scope of scrutiny compared to those that are followed by enforcement actions. The staggered nature of Blackburne et al.'s (2021b) [4] broader sample of all SEC investigations between 1999 and 2017 provides a powerful setting to infer causality; to our knowledge, this study is among the very few, if not the first, to examine the economic consequences of all SEC investigations on firms' future-oriented activities related to CSR. Unlike previous research that examines only material SEC investigations subject to enforcement (i.e., AAERs that represent only about 2% of all SEC investigations [12–15]), we find that all, even non-material, SEC investigations are associated with substantial declines in CSR performance. This finding appears to suggest that firms engage in CSR activities for opportunistic window dressing and reduce such activities when external pressures (e.g., SEC investigations, external monitoring by high-quality auditors) and concerns about CEO dismissal are high.

In our robustness tests, we address endogeneity concerns using dynamic analysis and propensity score matching, which lend further support to the main results. The lead-lag dynamic analysis reveals that the significantly negative relation between SEC investigations and CSR performance manifests mainly during the investigation period. The propensity matching methodology further shows that SEC investigations lead to a significant decrease in CSR for the firms under investigations.

There are several caveats about our results that should be noted. First, the dataset of SEC investigations is based on communications with the FOIA office of the SEC and lacks details about topics, dates of communications, and outcomes of the investigations, since the SEC does not comment on these specific details with outside stakeholders. Similar to Blackburne and Quinn (2023) [5], we are therefore unaware of the exact nature of the allegations pursued by the SEC during the investigation periods. Second, the SEC investigation dataset developed by Blackburne et al. (2021a; 2021b) [3,4] covers the period between 2000 and 2017 and no prior study has analyzed SEC investigations since then. Third, we cannot completely rule out concerns about potential endogeneity with respect to the observed relation between SEC investigations and CSR performance. For example, companies exhibiting strong CSR performance may encounter a lower likelihood of being subjected to SEC investigations due to the reputation-enhancing impact of CSR, which in turn diminishes the probability of SEC enforcement actions [12]. Furthermore, companies may be investigated due to unethical or wrongful behaviors but may also simultaneously improve their CSR practices as a part of their efforts to rebuild trust with stakeholders during the SEC investigation period.

Finally, although the MSCI KLD index is a widely used CSR rating system, there are some limitations to its use. First, the KLD index is designed for use with binary data, but CSR practices are often more complex and multidimensional in nature, thus making it difficult for real-world CSR practices to be represented in a binary fashion. Second, the KLD index assumes that all CSR practices are equally important or relevant to stakeholders; however, different industries and firms may have different priorities or expectations with respect to CSR practices. Lastly, given that measures of firms' corporate social performance tend to remain stable over time [19,83], it is also possible that the KLD index temporarily adjusts their CSR ratings during SEC investigations. To address these limitations, we encourage future research to further explore the relation between SEC investigations and CSR performance by developing different CSR measures (e.g., adherence to a reporting framework such as Global Reporting Initiatives and measures obtained from textual analysis) and analyzing investigations with known allegations.

**Author Contributions:** Conceptualization, X.L. and K.H.; methodology, X.L.; software, X.L.; formal analysis, X.L.; investigation, X.L.; resources, K.H.; writing—original draft preparation, J.-B.K.; writing—review and editing, K.H.; supervision, K.H.; funding acquisition, K.H. All authors have read and agreed to the published version of the manuscript.

**Funding:** This research was funded by the Social Sciences and Humanities Research Council of Canada, grant number 31-R832081.

**Institutional Review Board Statement:** Not applicable.

**Informed Consent Statement:** Not applicable.

**Data Availability Statement:** The data are available from the corresponding author upon request.

**Acknowledgments:** We thank Ray Zhang and Rafael Rogo for their comments and suggestions.

**Conflicts of Interest:** The authors declare no conflict of interest. The funders had no role in the design of the study; in the collection, analyses, or interpretation of data; in the writing of the manuscript; or in the decision to publish the results.

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
