# Peer review of "The Effect of the Security and Exchange Commission’s Investigations into Corporate Social Responsibility Performance"

_sustainability, doi:10.3390/su151914378_

Round 1

Reviewer 1 Report

Dear Authors,

The article offers a valuable investigation into the impact of the Security and Exchange Commission’s (SEC) investigations on firms' corporate social responsibility (CSR) performance. The methodology employed and the subsequent results provide noteworthy insights into this domain. However, there are a few suggestions and questions that may further refine the study:

Research Gaps:

While you have provided an in-depth exploration of the effects of SEC investigations on CSR performance, the manuscript does not clearly elucidate the research gaps that your study addresses. What makes your study distinct from prior research in this domain?

Is the interaction between SEC investigations and CSR performance a relatively unexplored topic? If there were previous studies, how does your study add to or challenge them? Further reading, https://doi.org/10.3390/su14106237 

Literature Review Matrix:

To better contextualize your study within the existing literature, I recommend the inclusion of a literature review matrix. This will help readers quickly identify the research gaps and contributions your study brings to the table.

The matrix can outline similar studies, their primary findings, methodologies, and how your study contrasts or aligns with them. Such a framework would provide a clearer picture of where your research fits within the broader academic discourse.

Theoretical Contribution:

The manuscript could benefit from a concise statement on how your findings build upon or deviate from existing theoretical frameworks on SEC investigations and CSR performance.

It would be insightful to know the theoretical implications of your findings. For instance, does your study challenge conventional wisdom or existing theoretical models? Does it call for a reevaluation of the interplay between regulatory investigations and corporate social activities?

In summary, your manuscript offers a compelling exploration of the relationship between SEC investigations and CSR performance. By addressing the above points, you can present a more rounded picture of the significance of your study in the grander scheme of the existing literature.

Best of luck!

Moderate editing of English language is required

Author Response

In this new version, we have conducted various tests suggested by the referees and expanded on the literature, so that our results make a more meaningful contribution. We believe that we have thoroughly addressed all concerns raised by the referees. We feel that the revised manuscript has improved considerably because of this constructive feedback.   

For ease of exposition and your reference, we have reproduced the referees’ comments in italic and then have placed our point-by-point responses directly below each comment. We have then highlighted the revised sections in yellow in the manuscript. We have had the paper professionally edited and prepared the manuscript in accordance with the journal’s editorial guidelines.

Reviewer 2 Report

The authors provided a sufficient and elegant description of the background of the topic herein along with the paper’s mission, which is quite relevant and timely as recent news coverage and accompanying concerns as the Security and Exchange Commission’s (SEC) investigationson firms’ corporate social responsibility (CSR) performance. Nonetheless, several suggestions will hopefully guide the authors in their efforts to improve the paper herein.

Comments

1)      What do the authors think about the potential drawback of calculating the overall CSR score (or performance) by subtracting the sum of CSR concerns from the sum of CSR strengths? The instrumental view of the stakeholder theory (Wood and Jones, 1995) states that not all stakeholders are equally important to all firms. Would not it be appropriate to utilize a measure that captures varying degrees of stakeholder importance on the part of firms subject to analysis herein?[1][295-309]

2)     The authors shall delete the variable “CSR_con” from Table 1 as it is a duplicate. [332-333]

3)     What is the rationale behind the inclusion of an indicator for loss as a control? In Table 3, the measure of profitability (i.e., variable “Profit”) has a minimum value of minus 1.14. How should we interpret it? [347-348]

4)     From Table 4, it appears that some results are somewhat unexpected about some of the control variables. Tobin’s q, which is usually utilized as a firm's forward-looking and market-based financial measure in CSR studies, appears to be negatively associated with overall  CSR performance and with CSR strengths, though not significantly. Slack resource's view of the firm predicts a positive association between the firm’s financial performance and its overall CSR performance.[2] Also, institutional ownership appears to be negatively and significantly associated with overall CSR score and CSR strengths, which is contradictory as institutional ownership promotes value creation by limiting opportunistic behaviors on the part of managers, thereby reducing the likelihood of agency problems occurring and improving governance. How do authors address the points raised thus far? [361-362].

5)     Why do authors not control for industry-specific effects of firms via industry-fixed effects as the importance of it is shown by McWilliams and Siegel (2000)[3]? [361-362].

6)     In Table 8, the robustness check was carried out by utilizing subdimensions of KLD’s 8 dimensions as dependent variables. It turns out that SEC investigations are associated negatively and significantly with governance and product dimensions and also being negatively associated with environment, employee relations, and human rights dimensions whereas positively associated with community and diversity dimensions (also significantly). Elaboration as to why these results turned out the way they did are lacking. The authors need to provide explanations as to why. Also, what do similar studies come up with? [450-466]

[1]  See, for instance, Akpinar, A., Jiang, Y., Gómez-Mejía, L. R., Berrone, P., & Walls, J. L. (2008). Strategic use of CSR as a signal for good management. Available at SSRN 1134505. https://papers.ssrn.com/sol3/papers.cfm?abstract_id=1134505, and also Choi, J. S., Kwak, Y. M., & Choe, C. (2010). Corporate social responsibility and corporate financial performance: Evidence from Korea. Australian journal of management, 35(3), 291-311. https://journals.sagepub.com/doi/10.1177/0312896210384681.

[2] See, for instance, a seminal paper: Waddock, S. A., & Graves, S. B. (1997). The corporate social performance–financial performance link. Strategic management journal, 18(4), 303-319. https://onlinelibrary.wiley.com/doi/abs/10.1002/(SICI)1097-0266(199704)18:4%3C303::AID-SMJ869%3E3.0.CO;2-G.

[3] Refer to McWilliams, A., & Siegel, D. (2000). Corporate social responsibility and financial performance: correlation or misspecification?. Strategic management journal21(5), 603-609. https://onlinelibrary.wiley.com/doi/10.1002/%28SICI%291097-0266%28200005%2921%3A5%3C603%3A%3AAID-SMJ101%3E3.0.CO%3B2-3.

Author Response

(The authors gave the same response as above.)

Round 2

Reviewer 1 Report

No further comments!

Minor editing of English language required.

Author Response

Thank you for reviewing our work and giving us an opportunity to revise and resubmit. We have fixed any remining minor edits and rephrased the content of the manuscript that was highlighted (i.e., content with more than 12 highlighted words in a row), as per the similarity report.